The blunted vascular endothelial growth factor-A (VEGF-A) response to high-altitude hypoxia and genetic variants in the promoter region of the VEGFA gene in Sherpa highlanders

Droma Yunden ydzmyp@shinshu-u.ac.jp 1
Hanaoka Masayuki masayuki@shinshu-u.ac.jp 1
Kinjo Takumi 1
Kobayashi Nobumitsu 1
Yasuo Masanori 1
Kitaguchi Yoshiaki 1
Ota Masao 2
1 The First Department of Medicine, Shinshu University School of Medicine , Matsumoto , Nagano , Japan
2 Department of Medicine, Division of Hepatology and Gastroenterology, Shinshu University School of Medicine , Matsumoto , Nagano , Japan
Mitsouras Katherine
Electronic publication date: 2022 Aug 17
Publication date: 2022
Volume: 10
Electronic Location ID: e13893
Received 2021 Jun 14; Accepted 2022 Jul 22
Copyright: ©2022 Droma et al.
Copyright year: 2022
Copyright holder: Droma et al.
License: This is an open access article distributed under the terms of the Creative Commons Attribution License, which permits unrestricted use, distribution, reproduction and adaptation in any medium and for any purpose provided that it is properly attributed. For attribution, the original author(s), title, publication source (PeerJ) and either DOI or URL of the article must be cited.
License URL: https://creativecommons.org/licenses/by/4.0/

Keywords: Genetic adaptation, High altitude, Hypoxia, Vascular endothelial growth factor, Sherpa highlanders

Funding: The authors received no funding for this work.

==============================
Background

Sherpa highlanders demonstrate extraordinary tolerance to hypoxia at high altitudes, which may be achieved by mechanisms promoting microcirculatory blood flow and capillary density at high altitudes for restoring oxygen supply to tissues. Vascular endothelial growth factors (VEGFs) are important signaling proteins involved in vasculogenesis and angiogenesis which are stimulated by hypoxia. We hypothesize that the VEGF-A, the major member of the VEGF family, and the gene encoding VEGF-A (VEGFA) play a part in the adaptation to high-altitude hypoxia in Sherpa highlanders.

Methods

Fifty-one Sherpa highlanders in Namche Bazaar village at a high altitude of 3,440 meters (m) above sea level and 76 non-Sherpa lowlanders in Kathmandu city at 1,300 m in Nepal were recruited for the study. Venous blood was sampled to obtain plasma and extract DNA from each subject. The plasma VEGF-A concentrations were measured and five single-nucleotide polymorphisms (SNPs, rs699947, rs833061, rs1570360, rs2010963, and rs3025039) in the VEGFA were genotyped. The VEGF-A levels and allelic frequencies of the SNPs were compared between the two populations.

Results

A significant difference in oxygen saturation (SpO2) was observed between the two ethnic groups locating at different elevations (93.7 ± 0.2% in Sherpas at 3,440 m vs. 96.7 ± 0.2% in non-Sherpas at 1,300 m, P < 0.05). The plasma VEGF-A concentration in the Sherpas at high altitude was on the same level as that in the non-Sherpas at low altitude (262.8 ± 17.9 pg/ml vs. 266.8 ± 21.8 pg/ml, P = 0.88). This result suggested that the plasma VEGF-A concentration in Sherpa highlanders was stable despite a high-altitude hypoxic stimulus and that therefore the Sherpas exhibited a phenotype of blunted response to hypoxic stress. Moreover, the allele frequencies of the SNPs rs699947, rs833061, and rs2010963 in the promoter region of the VEGFA were different between the Sherpa highlanders and non-Sherpa lowlanders (corrected P values = 3.30 ×10−5, 4.95 ×10−4, and 1.19 ×10−7, respectively).

Conclusions

Sherpa highlanders exhibited a blunted VEGF-A response to hypoxia at high altitudes, which was speculated to be associated with the distinctive genetic variations of the SNPs and haplotype in the promoter region of VEGFA in Sherpa highlanders.

Introduction

Atmospheric oxygen tension declines with altitude ascent. Generally, people from low altitudes exhibit oxygen insufficiency at high altitudes over 2,500 m (m) above sea level (Villafuerte & Corante, 2016). Historical sources indicate that the Sherpas in Nepal migrated from eastern Tibet to the Solu-Khumbu district in the Himalayan region in the 15th century (Gilbert-Kawai et al., 2014). They have permanently dwelled in the mountainous Himalayan region for more than 600 years (Gilbert-Kawai et al., 2014). Physiologically, Sherpa people exhibit extraordinary tolerance to hypoxia during the expeditions to extreme altitudes in the Himalayan region (Gilbert-Kawai et al., 2014). A series of studies have demonstrated that they adapt to and perform at high altitudes so well that they are rarely affected by acute or chronic mountain sicknesses due to various adaptations to hypoxia involving biological systems of hematology, respiration, cardiovascular system, and metabolism (Bhandari & Cavalleri, 2019; Gilbert-Kawai et al., 2014; Horscroft et al., 2017; Kinota et al., 2018). It has been also postulated that Sherpa highlanders are adapted to living in hypobaric hypoxia at high altitudes because they have a higher perfusion capacity of peripheral microcirculation, improved oxygen utilization efficiency, and more efficient tissue oxygen maintenance than the lowlanders (Davies et al., 2018; Gilbert-Kawai et al., 2017; Kayser et al., 1991). It has been widely hypothesized that genetic adaptation to high-altitude hypoxia, as a consequence of natural selection, may explain their adaptive phenotypes at high altitudes (Bhandari & Cavalleri, 2019; Gilbert-Kawai et al., 2014; Hanaoka et al., 2012).

VEGFs, a family of important growth factors, are key signaling proteins involved in both physiological and pathological angiogenesis through their influences on endothelial cell proliferation and migration (Holmes & Zachary, 2005). In mammals, the VEGF family comprises VEGF-A, VEGF-B, VEGF-C, and VEGF-D (Holmes & Zachary, 2005). VEGF-A is the major member of the VEGF family and is an endothelial cell-specific protein that has been implicated in multiple processes, including angiogenesis, vasculogenesis, and vascular endothelial cell division (Holmes & Zachary, 2005). The VEGF-A is highly expressed under hypoxia; its expression leads to endothelial cell proliferation and migration, thereby resulting in neovascularization induced by hypoxia in vivo and in vitro (Liu et al., 1995). Thus, we targeted VEGF-A and its polymorphisms in the VEGF-A gene (VEGFA) in the present study for investigating the hypothesis that the VEGF-A is involved with the adaptation to high altitude in Sherpa highlanders. Other members of the VEGF family were not included in the present study because there is little information available about the effects of hypoxia on these factors and further, the actions of VEGF-B, VEGF-C and VEGF-D primarily involve other systems, i.e., embryonic angiogenesis, lymphatic vascular endothelium, and lymphatic vasculature, respectively (Holmes & Zachary, 2005).

VEGF-A is encoded by the VEGFA gene on chromosome 6 (6p21.1). The gene consists of eight exons that are alternatively spliced to generate VEGF-A (Arcondéguy et al., 2013). Identification of the single nucleotide polymorphisms (SNPs) within the VEGFA demonstrated a correlation of the genetic variants with VEGF-A production (Watson et al., 2000). Therefore, possibly VEGFA gene variants may alter VEGFA expression and influence VEGFA-associated phenotypes (Appenzeller et al., 2006; Tomar, Malhotra & Sarkar, 2015).

The present study aimed to investigate the potential roles of the VEGF-A and the genetic variants of the VEGFA in adaptation to high-altitude hypoxia in Sherpa highlanders.

Materials & Methods

Ethics statement

The present study has been approved by the Ethics Committee of Shinshu University (Matsumoto, Japan; Permission numbers: 106) and the Nepal Health Research Council (Kathmandu, Nepal; Permission numbers: 2061-5-28 and Ref.182 061/62) and has therefore been performed following the ethical standards laid down in the Declaration of Helsinki. The protocol was individually explained to each Sherpa highlander and non-Sherpa lowlander, and each participant gave their informed consent written in Nepalese, by signature or by fingerprint if the subject was illiterate.

Study populations

The group of Sherpa highlanders comprised 51 Sherpas who resided in Namche Bazaar village (3,440 m) in the Solu-Khumbu region in Nepal. Namche Bazaar is the largest Sherpa village in Nepal residing with approximately 2,000 Sherpa inhabitants. It is a major gateway to Mount Everest and other famous peaks. The Sherpas voluntarily participated in this investigation. The Sherpa clan was identified with the Sherpa surname and confirmed by a senior native Sherpa. All enrolled Sherpas were born and permanently resided in Namche Bazaar and had no history of intermarriage with other ethnic groups. Medical interviews and physical examinations confirmed good health of them without chronic mountain sickness (CMS) and cardiopulmonary disorders. They did not travel to low altitudes before three months of the investigation. The oxygen saturation (SpO2) and pulse rate were measured and venous blood samples were taken at the high altitude of 3,440 m. The group of non-Sherpa lowlanders comprised 76 indigenous Nepalese university students, teachers, restaurant owners, and housewives living in Kathmandu (1,300 m) in Nepal. The Nepali identity was confirmed by their languages and social status (caste groups) based on Nepalese social demography with the help of our local staff. None of them had professional training in sports. All of them were in good health and had no mountaineering history before three months at the time of the investigation. In this group, the SpO2 and pulse rate were measured, and venous blood samples were obtained at the low altitude of 1,300 m.

The methods and protocols for subject recruitment and sample collection in the two groups as well as for the laboratory experiments are available as a collection in protocols.io: https://dx.doi.org/10.17504/protocols.io.byhdpt26.

Measurement of plasma VEGF-A concentrations

The plasma VEGF-A concentrations were measured using the quantitative sandwich ELISA kit (Human VEGF-A isoform Quantikine; R&D Systems, Minneapolis, MN) following the manufacturer’s instructions. Measurements were performed with samples in duplicate. The coefficients of variations of the measurements were 4.5% for intra-assay precision and 7.0% for inter-assay precision.

Selection of SNPs

The SNP rs3025039 in the 3′- untranslated region (UTR) of VEGFA was reported to be associated with altered plasma VEGF-A concentrations (Renner et al., 2000). Individuals carrying SNPs rs833061 in the promoter and rs2010963 in the 5′-UTR have been reported to be related to VEGFA promoter activity and responsiveness to biological stimuli, such as hypoxia (Stevens et al., 2003). Additionally, Prior et al. demonstrated that the VEGFA haplotype with three SNPs (rs699947, rs1570360, and rs2010963) was associated with enhanced VEGFA gene expression, and with maximal oxygen consumption in individuals before and after a standardized program of aerobic exercise training (Prior et al., 2006). Accordingly, SNPs rs699947, rs833061, rs1570360, rs2010963 in the promoter & 5′-UTR region, and rs3025039 in the 3′- UTR region of the VEGFA were genotyped in the Sherpa highlanders and non-Sherpa lowlanders (Table 1). Among these SNPs, the rs833061, rs2010963, and rs3025039 are quantitative trait loci (eQTLs) and contribute to variations in mRNA expression according to Human Genetic Variation Database (https://www.hgvd.genome.med.kyoto-u.ac.jp/).

Genotyping

The methods and protocols for DNA extractions and allele discriminations are available as a collection in protocols.io: https://dx.doi.org/10.17504/protocols.io.byhdpt26.

Statistical analysis

Continuous data were expressed as mean ± standard of error (SE). Student’s t-test for continuous data and the Chi-square test for categorical data were performed to detect significant differences between the two populations. Pearson correlation was applied to the assessment of the correlation between the plasma VEGF-A concentration and SpO2. The Hardy-Weinberg equilibrium (HWE) was evaluated for each SNP in the two populations. We estimated the strength of frequency of the major allele in Sherpa highlanders relative to non-Sherpa lowlanders by odds ratio (OR) with the approximate 95% confidence interval (CI). P values were corrected for multiple hypothesis tests with Bonferroni’s method. The corrected P (Pc) value of less than 0.05 was considered significant.

Table 1 Single-nucleotide polymorphisms of the VEGFA in this study.

SNP	Allele, Major/Minor	Position	Character and function	
rs699947	C/A	Promoter	Upstream intergenic (between genes) transcript variant	
rs833061*
(eQTL)	T/C	Promoter	Upstream transcript variant in the transcription factor binding site	
rs1570360	G/A	Promoter	Upstream transcript variant in the regulatory region	
rs2010963*
(eQTL)	C/G	5′-UTR	Upstream transcript variant	
rs3025039*
(eQTL)	C/T	3′-UTR	Downstream transcript variant	
Notes.

Abbreviations 3-′UTR 3′-untranslated region

5′-UTR 5′-untranslated region

SNPs single nucleotide polymorphisms

VEGFA gene encoding soluble vascular endothelial growth factor-A

A adenine

C cytosine

G guanine

T thymine

* eQTL: Expression quantitative trait loci that contribute to variation in mRNA expression according to Human Genetic Variation Database (https://www.hgvd.genome.med.kyoto-u.ac.jp/).

To examine the linkage disequilibrium (LD) of these SNPs in Sherpa highlanders and non-Sherpa lowlanders, the Haploview 4.2 software was used to derive the pairwise LD measurements and the logarithm of odds of LD (D′). The haplotype was estimated according to the D′values, based on the maximum likelihood to generate strong LD of the SNPs (Barrett et al., 2005).

Pair-wise genetic distance (FST) for SNPs rs699947, rs833061, rs1570360, rs2010963, and rs3025039 between high-altitude populations (Sherpa and Tibetan) and low-altitude populations (non-Sherpa Nepalese, East Asian, South Asian, and global population) were measured following (Weir & Cockerham, 1984). The genetic information of the SNPs in the high-altitude Tibetan population was sourced from Buroker et al. (2012), Buroker et al. (2013) and Buroker et al. (2017) and the genetic information in low-altitude populations (East Asian, South Asian, the whole global population) was from the 1000 Genomes Project (1000 Genomes Project, http://grch37.ensembl.org/Homo_sapiens/Info/Index, accessed May 2019).

Results

Blunted VEGF-A response to hypoxia in sherpas at high altitude

There were no significant differences in gender or age between the two groups (Table 2). None of the Sherpa highlanders complained of CMS symptoms at the high altitude. The SpO2 was significantly lower in Sherpas at 3,440 m (93.7 ± 0.2%) than non-Sherpas at 1,300 m (96.7 ± 0.2%, P = 6.43 × 10−17, Table 2). The plasma VEGF-A concentration in the Sherpas at high altitude was on the same level as that in the non-Sherpas at low altitude (262.8 ± 17.9 pg/ml vs. 266.8 ± 21.8 pg/ml, P = 0.88, Table 2). In addition, the plasma VEGF-A concentration was not significantly correlated with SpO2 in Sherpas at high altitude (r = 0.13, P > 0.05, Fig. 1A), whereas it was negatively correlated with SpO2 in non-Sherpas at low altitude (r = −0.27, P < 0.05, Fig. 1B).

Table 2 Phenotypes of Sherpa highlanders and non-Sherpa lowlanders.

Phenotypes	Sherpas
at high altitude	Non-Sherpas
at low altitude	P-value*	
No. subjects, n	51	76		
Males/Females, n (%)	21 (41.2)/30 (58.8)	40 (52.6)/36 (47.4)	0.27**	
Age, years	30.9 ± 1.1	29.9 ± 0.8	0.25	
Oxygen saturation, %	93.7 ± 0.2	96.7 ± 0.2	<0.001	
VEGF-A, pg/mL	262.8 ± 17.9	266.8 ± 21.8	0.88	
Pulse rate, bpm	82.4 ± 1.5	85.3 ± 1.8	0.21	
Notes.

Abbreviations bpm beats per minute

m meters above sea level

n number

VEGF-A vascular endothelial growth factor-A

Continuous data are expressed as mean ± standard of error (SE).

* P values were obtained by Student’s t-test, except ** by 2 × 2 contingency table.

Figure 1 Correlations of plasma VEGF-A concentrations with oxygen saturation (SpO2) in the Sherpas at high altitude and non-Sherpas at low altitude.

(A) Plasma VEGF-A concentration was not significantly correlated with the oxygen saturation (SpO2) in the Sherpas at high altitudes (r = 0.13, P > 0.05). (B) Plasma VEGF-A concentration was negatively correlated with the SpO2 in the non-Sherpas at low altitude (r = −0.27, P < 0.05).

VEGFA SNPs in sherpa highlanders versus non-sherpa lowlanders

The genotype distributions and allele frequencies of the SNPs all met the HWE in both groups. The genotype distributions of the SNPs in the promoter region (rs699947 and rs833061) and SNP in the 5′−UTR (rs2010963) differed between the two groups (Pc = 1.43 × 10−3, 2.48 × 10−3, and 1.04 × 10−7, respectively) as did their allele frequencies (Pc = 3.30 × 10−5, 4.95 × 10−4, and 1.19 × 10−7, respectively, Table 3). Particular attention was paid to the rs833061 and rs2010963 as these are eQTL loci in the promoter region and the 5′−UTR that could influence mRNA expressions and hence protein levels conventionally. On the other hand, genotype distributions of SNPs rs1570360 and rs3025039 did not differ between the two groups nor did their allele frequencies (Table 3). The major alleles of these SNPs (rs699947, rs833061, rs1570360, rs2010963, and rs3025039) were C, T, G, C, and C in the Sherpa highlanders; and C, T, G, G, and C in the non-Sherpa lowlanders correspondingly.

Table 3 Genotype distributions and allele frequencies of the SNPs in VEGFA in Sherpa highlanders (n = 51) versus non-Sherpa lowlanders (n = 76).

SNPs	Sherpa highlanders
n (frequency)	Non-Sherpa lowlanders
n (frequency)	Pc (genotype)*
Pc (allele)**	ORa
(95% CI)	
rs699947	
Genotype	
CC
CA
AA	35 (0.686)
14 (0.275)
2 (0.039)	25 (0.329)
39 (0.513)
12 (0.158)	1.43 × 10−3		
Allele	
C	84 (0.824)	89 (0.586)	3.30 × 10−5	3.30 (1.81–6 .04)	
A	18 (0.176)	63 (0.414)			
rs833061b	
Genotype	
TT
TC
CC	35 (0.686)
14 (0.275)
2 (0.039)	26 (0.342)
38(0.500)
12(0.158)	2.48 × 10−3		
Allele	
T
C	84 (0.824)
18 (0.176)	90 (0.592)
62 (0.408)	4.95 × 10−4	3.22 (1.76–5.88)	
rs1570360	
Genotype	
GG
GA
AA	41 (0.804)
9 (0.176)
1 (0.020)	45 (0.592)
29 (0.382)
2 (0.026)	0.20		
Allele	
G
A	91 (0.892)
11 (0.108)	119 (0.783)
33 (0.217)	0.10	2.29 (1.10–4.78)	
rs2010963b	
Genotype	
CC
CG
GG	21 (0.412)
26 (0.509)	4 (0.053)
39 (0.513)
33 (0.434)	1.04 × 10−7		
Allele	
C
G	68 (0.667)
34 (0.333)	47 (0.309)
105 (0.691)	1.19 × 10−7	4.47 (2.61–7.64)	
rs3025039b	
Genotype	
CC
CT
TT	35 (0.686)
15 (0.294)
1 (0.167)	57 (0.750)
18 (0.237)
1 (0.013)	3.65		
Allele	
C
T	85 (0.833)
17 (0.167)	132 (0.868)
20 (0.132)	2.15	0.76 (0.38–1.53)	
Notes.

Abbreviations Pc corrected P values

SNPs single-nucleotide polymorphisms

VEGFA gene encoding vascular endothelial growth factor-A

CI confidence interval

OR odds ratio

* Pc-genotype values were obtained by multiple hypothesis tests with Bonferroni’s method for the genotype distributions between the two groups (2 × 3 contingency table).

** Pc-allele values were obtained by multiple hypothesis tests for the allele frequencies between the two groups (2 × 2 contingency table).

a OR indicates the strength of the major allele in Sherpa highlanders relative to the non-Sherpa lowlanders by the allele frequency.

b eQTL: Expression quantitative trait loci that contribute to variation in mRNA expression according to Human Genetic Variation Database (https://www.hgvd.genome.med.kyoto-u.ac.jp/).

Haplotypes with the five SNPs in sherpa highlanders

Pair-wise LD analysis revealed that rs699947, rs833061, rs1570360, and rs2010963 SNPs in the promoter & 5-UTR region were strongly linked in both groups (Fig. 2). The haplotype comprising the Sherpa-major alleles (rs699947 C, rs833061 T, rs1570360 G, and rs2010963 C; C-T-G-C haplotype) exhibited a higher frequency in the Sherpa highlanders (0.667) than the non-Sherpa lowlanders (0.280, P = 3.1 × 10−5, Fig. 2, Table 4). Note that rs833061 and rs2010963 are the eQTL loci in the promoter region involved in the regulation of VEGFA mRNA expression. The VEGF-A concentrations in the Sherpas carrying the C-T-G-C haplotype at high altitude were on the same level as that in the non-Sherpas carrying this haplotype at the low altitude (240.9 ± 25.1 pg/ml vs. 227.9 ± 110.7 pg/ml, P > 0.05). However, there was no significant difference in the VEGF-A concentrations between the Sherpa highlanders with and without the C-T-G-C haplotype at high altitude (240.9 ± 25.1 pg/ml vs. 226.1 ± 30.9 pg/ml, P > 0.05), nor was that between the non-Sherpa lowlanders with and without this haplotype at low altitude (227.9 ± 110.7 pg/ml vs. 218.9 ± 24.3 pg/ml, P > 0.05).

Figure 2 Haplotypes with the SNPs of rs699947, rs833061 *, rs1570360, and rs2010963 * in the Sherpa highlanders and non-Sherpa lowlanders.

(A) The haplotype comprising rs699947 C, rs833061 T, rs1570360 G, and rs2010963 C (C-T-G-C haplotype) was prevalent in the Sherpa highlanders (frequency, 0.667). (B) The frequency of the C-T-G-C haplotype was 0.280 in the non-Sherpa lowlanders.* SNPs are expression quantitative trait loci (eQTL) that contribute to the variation in mRNA expression. A, adenine; C, cytosine; G, guanine; T, thymine.

Table 4 Haplotypes comprising the linked SNPs in Sherpa highlanders versus non-Sherpa lowlanders.

Haplotypes	Sherpa highlanders
(frequency)	Non-Sherpa lowlanders
(frequency)	P value*	

rs699947C-rs833061Ta-rs1570360G-rs2010963Ca	0.667	0.280	3.1 × 10 −5	
rs699947C-rs833061Ta-rs1570360G-rs2010963Ga	0.157	0.304	0.096	
rs699947A-rs833061Ca-rs1570360 A-rs2010963Ga	0.108	0.196	0.208	
rs699947A-rs833061Ca-rs1570360G-rs2010963Ga	0.069	0.110	0.554	
Notes.

Abbreviations SNPs single-nucleotide polymorphisms

* P values were obtained by Chi-square test with a 2 × 2 contingency table.

a eQTL: Expression quantitative trait loci that contribute to variation in expression levels of mRNA.

Genetic distance (FST) for each SNP between the high-altitude populations and low-altitude populations

As shown in Table 5, the FST values indicated that the five SNPs were very near between Sherpa and Tibetan high-altitude populations from 3.67 × 10−5 in rs699947 to 0.011 in rs2010963. Except for SNP rs3025039 which showed quite near the genetic distance between high- and low-altitude populations, the other four SNPs rs699947, rs833061, rs1570360, and rs2010963 in the C-T-G-C haplotype exhibited considerably separated genetic distance between the high-altitude populations (Sherpa and Tibetan) and low-altitude populations (non-Sherpa Nepalese, East Asian, South Asian, and global population). Among these four SNPs, SNP rs2010963 was the most separated genetic variation between Sherpa highlanders and other populations (Table 5).

Table 5 Pair-wise genetic distance (FST) for VEGFA gene SNPs rs699947, rs833061, rs1570360, rs2010963, and rs3025039 between the high-altitude populations and low-altitude populationsa.

SNPs	High-altitude populations	Low-altitude populations	
	Sherpa &
Tibetan	S herpa	non-Sherpa Nepalese	East Asian	South Asian	Global population	
rs699947	Sherpa	0	0.068	0.016	0.059	0.029	
Tibetan	3.67 × 10−5	0.071	0.017	0.062	0.031	
rs833061*	Sherpa	no data	0.065	0.016	0.060	0.047	
Tibetan	no data	no data	no data	no data	no data	
rs1570360	Sherpa	0	0.022	0.012	0.037	0.013	
Tibetan	6.45 × 10−4	0.030	0.017	0.047	0.019	
rs2010963*	Sherpa	0	0.128	0.067	0.175	0.116	
Tibetan	0.011	0.067	0.025	0.104	0.059	
rs3025039*	Sherpa	0	0.002	1.58 × 10−5	0.007	0.002	
Tibetan	0.009	0.002	0.009	9.33 × 10−5	0.002	
Notes.

Abbreviations SNPs single-nucleotide polymorphisms

a The genetic information of Tibetans was obtained from Buroker et al. (2012); Buroker et al. (2013); Buroker et al. (2017). The genetic information of the populations in East Asia and South Asia, as well as the whole global population were obtained from the 1000 Genomes Project (1000 Genomes Project, http://grch37.ensembl.org/Homo_sapiens/Info/Index, accessed May 2019).

The pair-wise FST values for the SNPs between the high- and low-altitude populations were measured following Weir & Cockerham (1984).

High-altitude populations: Sherpa and Tibetan; Low-altitude populations: non-Sherpa Nepalese, East Asian, South Asian, and global population.

* eQTL: Expression quantitative trait loci that contribute to variation in mRNA expression according to Human Genetic Variation Database (https://www.hgvd.genome.med.kyoto-u.ac.jp/).

Discussion

The most remarkable finding in the present study was that the VEGF-A concentration did not increase in Sherpa highlanders with the stimulus of high-altitude hypoxia, indicating that the plasma VEGF-A in Sherpas did not respond to high-altitude hypoxia and thus suggested a blunted response to hypoxia in Sherpa highlanders. Moreover, this blunted VEGF-A response to hypoxia was accompanied by differences in the genetic variants in the promoter region of the VEGFA gene between Sherpa highlanders and non-Sherpa lowlanders.

Hypoxia is an important regulator of blood vessel tone and structure, and a potent stimulus for vasculogenesis and angiogenesis (Liu et al., 1995). VEGF-A production is mainly stimulated by hypoxia and promotes the proliferation of vascular smooth muscle cells and endothelial cells under hypoxic conditions and thereby potentially vasculogenesis and angiogenesis (Moens et al., 2014; Namiki et al., 1995). Hypoxia-stimulated vasculogenesis and angiogenesis may increase microcirculatory blood flow and capillary density, thus restoring the oxygen supply in tissues exposed to hypoxia to compensate for oxygen insufficiency (Carmeliet, 2000). High-altitude medicine studies have revealed increased circulating VEGF-A concentrations which were negatively correlated with SpO2 (Ge et al., 2011; Tissot Van Patot et al., 2005; Walter et al., 2001; Zhang et al., 2018). Thus, we anticipated that the VEGF-A concentration would increase in Sherpas at high altitudes compared with non-Sherpa lowlanders, perhaps resulting in an increase of capillary density in tissues under high-altitude hypoxia. Indeed, compared to lowlanders, Sherpa highlanders exhibited a higher number of capillaries per square millimeter muscle cross-sectional area, which would be expected to improve the efficient and effective diffusion of oxygen to muscles (Kayser et al., 1991). Moreover, Sherpas exhibited higher sublingual capillary density and faster microcirculatory flow per time and volume of tissues compared to lowlanders at 5,300 m (Gilbert-Kawai et al., 2017). All these studies suggested that the VEGF-A and the associated microcirculatory physiology could play important roles in the process of adaptation to high-altitude hypoxia in Sherpa highlanders. Unexpectedly, the present results showed that the plasma VEGF-A levels in Sherpa highlanders were on the same level as that in the non-Sherpas at low altitude despite a high-altitude hypoxic stimulus, and therefore exhibited a phenotype of blunted response to hypoxia stress.

VEGF-A may also play a role in maladaptation to high altitudes as excessive production of VEGF-A at high altitudes could contribute to the pathophysiological formation of abnormal blood vessels, pulmonary vessel remodeling, vascular smooth muscle cell proliferation (Appenzeller et al., 2006; Ge et al., 2011) and distal vasculogenesis (Ma et al., 2015) that are potential contributors to CMS, a syndrome of maladaptation to high altitudes (Villafuerte & Corante, 2016). Plasma VEGF-A concentrations were negatively correlated with SpO2 at high altitudes and elevated in CMS patients (Ge et al., 2011). Thus, we propose that the blunted VEGF-A response to hypoxia may contribute to preventing over-production of VEGF-A, pathophysiological vasculogenesis, and angiogenesis in Sherpas at high altitudes. A blunting of hypoxic response has also been seen in healthy Tibetans at high altitudes. For example, hemoglobin (Hb) concentrations were relatively low in Tibetans at high altitudes and within the normal sea-level range (Beall et al., 1998), and the same phenotype was also observed in pulmonary arterial pressures in Tibetans at high altitudes (Groves et al., 1993). Pulmonary artery pressure and Hb are sensitive to hypoxia stress and hypoxia-induced pulmonary hypertension and polycythemia are typical phenotypes of maladaptation to high-altitude hypoxia (Appenzeller et al., 2006; Ge et al., 2011). Therefore, blunted responses to hypoxia may serve to suppress the over-responses of Hb, pulmonary vasoconstriction, and vasculogenesis seen in maladaptive syndromes or non-adapted populations.

In this study, the blunted VEGF-A levels and VEGFA gene variants in Sherpa highlanders (Namche Bazaar) were compared to non-Sherpa lowlanders (Kathmandu) in Nepal. The Namche Bazaar and Kathmandu are geographically close with 200 kilometers of direct distance. Geographic distance is an excellent predictor of the neutral genetic diversity of human populations, which explains 85% of the observed variance on a worldwide scale (Prugnolle, Manica & Balloux, 2005), and geography is a better determinant of human genetic differentiation than ethnicity (Manica, Prugnolle & Balloux, 2005). In addition, a recent study found a high level of genetic affinity of Sherpas to those of South Asian ancestry, probably due to bi-directional gene flow (Zhang et al., 2017). And Sherpas exhibit genetic affinity to some Nepalese tribes (Cole et al., 2017). Under these circumstances, the subjects of non-Sherpa Nepalese in Kathmandu were chosen as the lowlanders offering a contrasting counterpart to Sherpa highlanders for the present study. Our study suffered from several weaknesses. One of the limitations of this study is that there were no data for plasma VEGF-A concentrations in Sherpas at low altitudes nor were studies conducted in non-Sherpas at high altitudes. Such studies are needed but it was not possible for the present study to carry out such group movements due to logistical and other constraints. Nonetheless, several studies have demonstrated that the VEGF-A levels are increased by hypoxia stress (Ge et al., 2011; Tissot Van Patot et al., 2005; Walter et al., 2001; Zhang et al., 2018). Another limitation is that it lacked measurements of angiogenesis and vasculogenesis in Sherpas at high altitudes due to the lack of requisite analytic equipment and the limitations of performing invasive studies in Namche village. In addition, the small sample sizes made the statistical power relatively weak. However, the strength of our study is that it was the first study to measure plasma VEGF-A concentrations in combination with VEGFA gene variants in Sherpas at a high altitude.

We genotyped the rs699947, rs833061, rs1570360, rs2010963, and rs3025039 SNPs for VEGFA in the Sherpa highlanders and non-Sherpa lowlanders. The results revealed genetic divergences in the frequencies of the rs699947, rs833061, rs2010963 SNPs (Table 3) and in the haplotype (rs699947 C- rs833061 T- rs1570360 G- rs2010963 C, C-T-G-C) between the two populations (Table 4). Given that the VEGF-A levels were increased in patients with CMS (Ge et al., 2011) but blunted in Sherpas with the C-T-G-C haplotype (Table 4), the C-T-G-C haplotype may have contributed to the blunted VEGF-A response in Sherpas. Further, we speculate that the rs833061 and rs2010963 eQTL loci in the haplotype may downregulate the mRNA expression and hence VEGF-A levels. Further study is required to determine whether such effects occur and can help to explain the blunted VEGF-A response to hypoxia in Sherpa highlanders.

Tibetan population is the ancestry of Sherpa people in the Himalayan region and they have been residing on Tibetan Plateau for thousands of years and developed unique characteristics to adapt to high altitudes (Bhandari et al., 2015). Sherpas and Tibetans possess common adaptive traits for adaptation to high altitudes (Bhandari et al., 2015). The allele frequencies of rs699947, rs1570360, rs2010963, and rs3025039 are at the same levels between Sherpas and Tibetans (Table 6) according to the genetic information of these SNPs in Tibetans (Buroker et al., 2012; Buroker et al., 2013; Buroker et al., 2017). These genetic variants of VEGFA are likely involved with the adaptation to high-altitude hypoxia in Himalayan highlanders. While on the other hand, the allele frequencies of rs699947, rs833061, rs1570360, and rs2010963 in Sherpa highlanders differ from those in various low-altitude populations in East Asia and South Asia, as well as of the whole global population (Fig. 3) based on the genetic information from the 1000 Genomes Project (1000 Genomes Project, http://grch37.ensembl.org/Homo_sapiens/Info/Index, accessed May 2019). Genes with strong frequency differences between populations are potential targets of natural selection (Yi et al., 2010). The population-specific allele frequency changes were estimated by including two high-altitude populations (Sherpa and Tibetan) and four low-altitude populations (non-Sherpa Nepalese, East Asian, South Asian, and global population). The pair-wise FST values on genetic distance between the high- and low-altitude populations (Table 5) inferred an estimation that the allelic frequency alterations in rs699947, rs833061, rs1570360, and rs2010963 of the C-T-G-C haplotype in the VEGFA have occurred in the Sherpa and Tibetan populations since they diverged from the low-altitude populations. This estimation suggests that the divergencies in the variants of the VEGFA gene in Himalayan highlanders likely resulted from a process of natural selection for genetic adaptation to high altitudes.

Table 6 Allele frequencies of the SNPs rs699947, rs833061, rs1570360, rs2010963, and rs3025039 in the high-altitude populations of Sherpa and Tibetan*.

SNPs	Populations	Number of subjects	Major allele
(frequency)	Minor allele
(frequency)	P a	
			C	A		
rs699947 (C/A)
promoter	Sherpa
Tibetan	51
32	0.824
0.828	0.176
0.172	0.94	
			G	A		
rs1570360 (G/A)
promoter	Sherpa
Tibetan	51
27	0.892
0.907	0.108
0.093	0.76	
			C	G		
rs2010963 (C/G)b
5′-UTR	Sherpa
Tibetan	51
30	0.667
0.567	0.333
0.433	0.21	
			C	T		
rs3025039 (C/T)b
3′-UTR	Sherpa
Tibetan	51
29	0.833
0.897	0.167
0.103	0.27	
Notes.

Abbreviations SNPs single-nucleotide polymorphisms

UTR untranslated region

* The genetic information in Tibetans was obtained from Buroker et al. (2012); Buroker et al. (2013); Buroker et al. (2017).

a P values were obtained by Chi-square test with a 2 × 2 contingency table.

b eQTL, Expression quantitative trait loci that that contribute to variation in expression levels of mRNA.

Figure 3 Genetic divergences of the five SNPs in high-altitude populations (Sherpa and Tibetan) versus low-altitude populations (non-Sherpa Nepalese, East Asian, South Asian, and the whole global population).

The genetic information of Tibetans was obtained from Buroker et al. (2012), Buroker et al. (2013) and Buroker et al. (2017). The genetic information of the populations in East Asia and South Asia, as well as the whole global population, were obtained from the 1000 Genomes Project (1000 Genomes Project, http://grch37.ensembl.org/Homo_sapiens/Info/Index, accessed May 2019).Yellow bars: major alleles for the high-altitude populations (Sherpa and Tibetan); Green bars: major alleles for the low-altitude populations (non-Sherpa Nepalese, East Asian, South Asian, and the whole global population); Light green bars: minor alleles for all populations; Numbers on the bars: allele frequencies. *SNPs are expression quantitative trait loci (eQTL) that contribute to the variation in mRNA expression. A, adenine; C, cytosine; G, guanine; T, thymine.

The eQTL loci of rs833061 and rs2010963 in the promoter region of VEGFA involve in the VEGFA mRNA expression, though many factors have been proposed to influence mRNA–protein correlation including mRNA stability and protein degradation etc. (de Sousa Abreu et al., 2009). Genetic variations of VEGFA are related to hypoxic response likely through targeting transcription factor binding sites (TFBS) in the promoter region in which hypoxia-responsive elements (HREs) mediate transcriptional mechanism to induce hypoxia-induced gene expression (Forsythe et al., 1996; Pages & Pouyssegur, 2005). The DNA sequence variation in the promoter region of VEGFA impacts the gene expression and maximal oxygen consumption (Prior et al., 2006). The SNPs rs699947 and rs2010963 were associated with altered VEGFA promoter activity and responded to hypoxia exposure (Liu et al., 1995). Moreover, the SNP rs699947 provides a binding location for the hypoxia-induced factor (HIF)-alpha protein dimer to attach to the VEGFA promoter and regulate the gene (Forsythe et al., 1996). These potentially interesting pieces of evidence together with the present results support a speculative mechanistic model as shown in Fig. 4. The genetic variations of the VEGFA in Sherpa highlanders down-regulate the VEGFA mRNA expression levels by interacting with other hypoxia-associated genes (such as HIF) (Bigham & Lee, 2014), resulting in high-altitude adaptive VEGF-A which is blunted response to hypoxia and beneficial to adequate oxygen supply in tissues for adaptation to high altitudes (Fig. 4).

Figure 4 The speculative mechanistic model of the vascular endothelial growth factor-A (VEGF-A) and the single nucleotide polymorphisms (SNPs) in the promoter region of the VEGFA gene in adaptation to high-altitude hypoxia in the Sherpa highlanders.

The genetic variations of the VEGFA in Sherpa highlanders down-regulate the VEGFA mRNA expression levels by interacting with other hypoxia-associated genes (such as HIF), resulting in high-altitude adaptive VEGF-A which is blunted response to hypoxia and beneficial to adequate oxygen supply in tissues for adaptation to high altitudes. The red solid represents promoter and 5′-UTR; the black solid represents exon; the blank represents intron. * SNPs are expression quantitative trait loci (eQTL) that contribute to the variation in mRNA expression. **The VEGFA gene is one of the downstream genes of the HIF pathway involving angiogenesis. Major alleles (orange)/Minor alleles (blue) of the SNPs rs699947, rs833061, rs1570360, and rs2010963 of the VEGFA in the Sherpa highlanders. Abbreviations: HIF, hypoxia-inducible factor; HRS, hypoxia-responsive element; UTR, untranslated region.

Conclusions

In summary, VEGF-A levels in Sherpa highlanders exhibited a blunted response to hypoxia at high altitudes, which was accompanied by the divergencies in SNPs in the promoter region of VEGFA. Natural selection by high-altitude hypoxia may have favored a VEGFA haplotype conferring an adaptive advantage to the Sherpa population. Further study, with adequate controls for ancestry and altitude, is necessary to identify the mechanisms by which such SNPs may be acting to affect protein levels and phenotypic correlates, as well as to identify possible interactions between VEGFA with other hypoxia-associated genes.

Supplemental Information

Supplemental Information 1 VEGF phenotypes in the two groups

Raw data for phenotypes of Sherpa highlanders and non-Sherpa lowlanders (Table 2).

Click here for additional data file.

We are grateful to all the Sherpa highlanders and non-Sherpa lowlanders for their kind participation in this study. We thank Doctors: Buddha Basnyat, Amit Arjyal, Pritam Neupane, Anil Pandit, and Dependra Sharma for their assistance during sample collections in the Sherpa village and Kathmandu in Nepal. We appreciated the cooperation from the Nepal Health Research Council (Katmandu, Nepal).

Additional Information and Declarations

Competing Interests

Author Contributions

Human Ethics

Field Study Permissions

Data Availability

The authors declare there are no competing interests.

Yunden Droma conceived and designed the experiments, performed the experiments, analyzed the data, prepared figures and/or tables, authored or reviewed drafts of the article, and approved the final draft.

Masayuki Hanaoka conceived and designed the experiments, authored or reviewed drafts of the article, and approved the final draft.

Takumi Kinjo analyzed the data, prepared figures and/or tables, authored or reviewed drafts of the article, and approved the final draft.

Nobumitsu Kobayashi performed the experiments, analyzed the data, authored or reviewed drafts of the article, and approved the final draft.

Masanori Yasuo analyzed the data, authored or reviewed drafts of the article, and approved the final draft.

Yoshiaki Kitaguchi analyzed the data, authored or reviewed drafts of the article, and approved the final draft.

Masao Ota conceived and designed the experiments, analyzed the data, prepared figures and/or tables, authored or reviewed drafts of the article, and approved the final draft.

The following information was supplied relating to ethical approvals (i.e., approving body and any reference numbers):

The present study has been approved by the Ethics Committee of Shinshu University (Matsumoto, Japan; Permission numbers: 106) and the Nepal Health Research Council (Kathmandu, Nepal; Permission numbers: 2061-5-28) and has therefore been performed in accordance with the ethical standards laid down in the Declaration of Helsinki.

The following information was supplied relating to field study approvals (i.e., approving body and any reference numbers):

Fieldwork was approved by the Nepal Health Research Council (Kathmandu, Nepal) (2061-5-28 and Ref.182 061/62).

The following information was supplied regarding data availability:

Raw data for phenotypes of Sherpa highlanders and non-Sherpa lowlanders are available in the Supplemental Files.

Raw data concerning genotypes is available at Zenodo:

Yunden Droma. (2021). The genotypes of variations of VEGFA in Sherpa highlanders and non-Sherpa lowlanders [Data set]. Zenodo. https://doi.org/10.5281/zenodo.4816702.

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
