# Peer review of "The blunted vascular endothelial growth factor-A (VEGF-A) response to high-altitude hypoxia and genetic variants in the promoter region of the VEGFA gene in Sherpa highlanders"

_PeerJ, doi:10.7717/peerj.13893_

## Round 0.1 · original submission · Major Revisions

Your manuscript was considered novel, interesting and valuable by the reviewers. However, they raised some important issues that need to be addressed.

There was an overarching concern about your conclusions not being supported by your findings. More, specifically, there was a concern about the confounding effect of altitude/hypoxia and genetic background, which affects the validity of your conclusion that Sherpas have a decreased concentration of VEGF-A at altitude due to their genetic background.

The reviewers would also like to see a functional investigation of the effect of the major SNPs/SNVs in your study on VEGF-A expression or, alternatively, including any publicly available eQTL data on these SNPs/SNVs.

Additional issues that were raised were performing a formal selection test on your data and a formal association test between the genetic variants and VEGF-A levels, as well as including a discussion of metabolic adaptations to high altitudes in Sherpas and considering alternative hypotheses to explain your observation of the lack of difference in VEGF-A between Sherpas and lowlanders being due to decreased VEGF-A expression.

Please, submit a detailed rebuttal which shows where and how you have taken all comments and suggestions into consideration. If you do not agree with some of the reviewers’ comments or suggestions, please explain why. Your rebuttal will be critical in making a final decision on your manuscript. Please, note also that your revised version may enter a new round of review by the same or by different reviewers. Therefore, I cannot guarantee that your manuscript will eventually be accepted.

Reviewer 1 ·

Basic reporting

In this manuscript, Droma et al show that the VEGF-A expression in Sherpas at high altitude was tolerant to hypoxia and associated with distinctive genetic variants.
The manuscript has been written in clean and unambiguous, professional English. Proper background/context and literature references have been provided. The methods have been explained in great details. I found the study unique and very interesting. I have only few minor comments for the authors.

Experimental design

The authors pose simple, straightforward questions and their experiments have been designed accurately in a manner that directly test their hypotheses. Method section is very well written.

Validity of the findings

The authors' conclusions are accurate, logical and in accordance to the findings of their experiments. All data have been provided and the data appear statistically sound with proper controls.

Additional comments

I have only few minor comments:
1. Did the authors looka at other isoforms of VEGF-b,c,d, and if not they should mention these in the introduction and provide a rationale regarding why they chose to look only at VEGF-A.
2. In several parts of the manuscript, the hyphen is missing in VEGFA (esp in methods section). Please make them consistent with the rest of the paper.
3. Line 173 it says there is no significant difference in gender ratio between the 2 groups (table 2). That is strange because the ratio is quite different 21/30 compared to 40/36. Can the authors please explain this?
4. Line 212 there is a typo - should be "upregulated" not "upregulates".
5. Figure 4 legend line 2 should be "East" instead of "Esat"
6. In line 232, please provide the full form of CMS.
7. In line 245, mention "table 4".
8. Please rewrite line 272 " The SNPs rs699947...." so that it appears cohesive.
9. Line 281, should be "VEGF-A", currently its "VEGE-A".

·

Basic reporting

The authors report on an investigation of VEGFA genotype and circulating VEGFA levels in lowlander Nepalese and high-altitude dwelling Sherpas. The language used is clear and unambiguous throughout, though there are some minor grammatical errors throughout which suggests that the manuscript would benefit from proofreading by a native English speaker. There are too many examples for me to list, but for example, in the concluding section of the abstract:

"The VEGF-A in Sherpas at high altitude exhibited a phenotype that was tolerant
to high-altitude hypoxia."

This does not quite make sense, as it would be the Sherpas who exhibit a phenotype, rather than the protein. The protein (or rather, expression of it) is a feature of that phenotype.

The literature review is well conducted, particularly around VEGFA - the main topic of the paper. My one comment would be on this statement:

"In addition to adaptation to hypoxia majorly by mechanisms through hematology, pulmonary and cardiovascular systems (Bhandari & Cavalleri 2019; Gilbert-Kawai et al. 2014)..."

The authors should also note that Sherpas exhibit metabolic adaptation to high-altitude, including suppression of fatty acid oxidation, in association with selection on the PPARA gene. This is likely to be a key component of the adaptation and the original work (from our group and Tatum Simonson and others) is cited in the Bhandari review.

Experimental design

The experimental design is straightforward enough and appropriate. The Methods section is overly long, however, and the authors should aim to shorten the section on Study Populations (lines 90-130). There is a lot of unnecessary background information here, which is not appropriate for this section. In particular, the material from line 92-97 describing the authors' trek to Namche Bazaar is not really necessary. The material from 120-130 is better suited to the Introduction or Discussion.

Validity of the findings

The results are presented in an adequate way and the statistical methods appear to be sound. The results are straightforward observations, and the measures appear to have been conducted well.

My major concern is with the conclusion that the VEGFA phenotype in Sherpas demonstrates resilience against hypoxic stress. There is no evidence that the Sherpas are hypoxically stressed, and neither would one expect there to be, given their adaptation to this altitude is clear. I can understand that the authors are attempting to reconcile the finding of no difference in VEGFA levels in plasma, with the clear finding of genetic selection, but the data does not support this conclusion. The critical experiment would be to take lowlander Sherpas and non-Sherpas to high altitude and monitor changes in VEGFA expression in individuals as they ascend. The authors discuss this as a limitation in their manuscript, but for me this is more than a limitation and would be necessary to support the conclusion of a phenotype that is not associated with increased VEGFA at altitude.

In our previous work, we measured muscle VEGFA RNA, and found that it increased in both lowlander Europeans and lowland-dwelling Sherpas upon ascent to Everest Base Camp (Horscroft et al. PNAS 2017, Fig 1D). An alternative explanation for the authors' results, compatible with previous findings, would be that the VEGFA response to altitude happens with acute exposure or during development, and is under negative feedback control. In the high-altitude dwelling Sherpa, this response has taken place and along with other features of their adaptation, has mitigated against the tissue hypoxia, such that VEGFA expression is no longer elevated.

Finally, as the authors do state, VEGFA is under transcriptional control by the hypoxia inducible factor pathway. There is strong genetic selection in Tibetan populations on genes that encode components of this pathways, chiefly EPAS1 and EGLN1. The authors should note that these variants might also influence VEGFA expression, in addition to the VEGFA variants themselves.

Whilst the authors may well be correct that the lack of difference in VEGFA between Sherpas and lowlanders may represent a suppression of expression associated with the genotype, they cannot definitively conclude this and should alter their manuscript to include these alternative hypotheses.

Additional comments

I enjoyed this manuscript, and think there are interesting findings here. I hope my comments are useful to you.

Andrew Murray, University of Cambridge, UK

Reviewer 3 ·

Basic reporting

1. Article can be improved by use of clear and grammatically correct English throughout. Multiple sentences have ambiguous wording, spelling, and syntax. For example:

Line 51: “Sherpa people are originally from Tibetan highlanders” – is confusing, it’s unclear if this is a typo and the authors meant ‘originally from the Tibetan highlands’ or if they mean that the Sherpa people share a recent common ancestor with Tibetan highlanders as stated later in the text

Lines 55-60: “In addition to adaptation to hypoxia majorly by mechanisms through hematology, pulmonary and cardiovascular systems (Bhandari & Cavalleri 2019; Gilbert-Kawai et al. 2014), the adaptation to high-altitude hypoxia in Sherpa highlanders is partly by an additional mechanism that promotes increases in microcirculatory blood flow and capillary density at high altitudes for restoring oxygen supply to tissues (Davies et al. 2018; Gilbert-Kawai et al. 2017; Kayser et al. 1991). – The syntax here is awkward and difficult to follow such as ‘majorly by mechanisms’, and the sentence is long and loses clarity, I would recommend breaking this up into multiple sentences.

Line 111 and 112 uses the term precipitation, which I believe means ‘precipitate’ or ‘pellet’, the use of this term here is confusing with regard to pelleted blood cells.

Figure 3A, "of them carrying the C-T-G-C haplotype" should be "of those"

Line 205, “in” is italicized and should not be.

Line 292-292: "Natural selection by the high-altitude hypoxia would have favored VEGFA to adaptive advantage for Sherpa people dwelling at high altitudes." - "would have favored VEGFA to adaptive advantage" seems like a word is missing. Do the authors mean that a specific VEGFA haplotype was favored by natural selection amongst high-altitude Sherpa populations?

2. Literature and references appear to be sufficient for the field.

3. Figure 1 and Figure 3 appear redundant. Figure 1 shows the overall data from section 1 of the results which reports no difference in VEGF-A concentration between Sherpas and non-Sherpas as well as differing correlations between VEGF-A and SpO2 between the two populations. Figure 3 and section three of the results do the same thing, but limiting it to the subset of each group carrying the C-T-G-C haplotype. Because the results essentially do not differ whether you separate out this haplotype or not, its unclear with this adds to the paper.

4. Article is self-contained and well organized. The data presented are relevant to the hypothesis, but in my opinion do not confidently support the conclusions.

Experimental design

1. A primary design flaw of this investigation is that genetic background (Sherpa vs non-Sherpa) and altitude/hypoxia (1300m vs 3440m) are entirely confounded. It is difficult to simultaneously conclude that the Sherpa have a lower than expected VEGF-A concentration at altitude and that it is due to their genetic background while having no data on either Sherpas at low altitude or non-Sherpas at high altitude. If the Sherpas baseline VEGF-A concentration is lower at 1300m than the non-Sherpa, it may be that the effect of hypoxia on VEGF-A is no different between them, but merely that their baseline is different due to genetic background which may or may not have to do with the gene itself or hypoxic response. The authors briefly address this drawback in the Discussion, but it may be worth drawing more attention to or explaining in more depth why their results are still likely valid regardless.

2. The authors do a good job of defining how and why their research question is meaningful and fills a gap in knowledge, while acknowledging that they do not have the optimal samples to full address it.

3. The authors provide ample support for rigorous ethical standards in their investigation.

4. The findings are relatively short, with a very brief results section. The only supplemental data provided are VEGF-A levels as well as SpO2, BPM, age and sex of participants. I believed the design could be improved by more rigorous investigation including the following:

For the major SNPs identified as potentially important in this study, are the alleles identified associated with levels of VEGF transcription. All alleles appear to be non-coding, so the assumption would be that they alter regulatory activity. This is briefly discussed in the discussion, but that reports only that: “Functional analysis of the VEGFA 5ʹ-UTR in hypoxic Hep3B cells has revealed that the rs699947 SNP provides a binding location for the hypoxia-induced factor (HIF)-alpha protein dimer to attach to the VEGFA promoter and regulate the gene (Forsythe et al. 1996). The SNPs rs699947 and rs2010963 were associated with significantly altered VEGFA promoter activity and responded to hypoxia exposure (Liu et al. 1995).” (lines 274-278). This is unclear for several reasons, is rs699947 disrupting HIF binding activity, or creating a previously non-existent HIF-binding site, and if so do the alleles more common in Sherpas agree with the directionality of the binding site disruption? Which alleles at rs699947 and rs2010963 are associated with increased and decreased promoter activity and what effect does hypoxia have on them? Adding in functional experiments such as enhancer assays or ChiP could better answer these questions and improve upon the 25 year old original research on these SNPs. Alternately, since the alleles appear relatively common, it is possible larger consortia have publicly available information on eQTL status of these SNPs. Minimally, reporting the direction of effect for each allele at these SNPs either previously reported or identified in your data is essential.

An analysis of ancestry using admixture or similar of the two test populations, in particular relating to the non-Sherpa populations who might be expected to be more admixed that the Sherpa populations

Performing or data mining selection scans on these alleles. The Conclusion implies that there may be signatures of selection on the SNPs, but does not report a formal test. Given the multitude of large selection scans on Tibetan populations, and the genetic similarities between Sherpa and Tibetans, it is possible this has already been done or could be performed using available data and imputation.

Validity of the findings

1. The primary conclusions of this article - “the VEGF-A in Sherpa highlanders is tolerant to hypoxia stress at high altitudes. This hypoxia-tolerant VEGF-A is likely associated with the significant genetic divergences in the VEGFA in Sherpa highlanders. Natural selection by the high-altitude hypoxia would have favored VEGFA to adaptive advantage for Sherpa people dwelling at high altitudes.” – do not seem well supported by the data. While the data may suggest this hypothesis, it certainly does not conclusively prove it. As stated above, genetic background and altitude/hypoxia are completely confounded in the experimental design, so there’s no way to conclusively test that the Sherpa have blunted or ‘hypoxia-tolerant’ expression of the VEGF-A. Moreover, no quantitative tests of QTLs or positive selection are reported to demonstrate with statistical significance that the alleles reported are associated with altered VEGFA expression or that these alleles demonstrate signatures of selection.

2. The conclusions are well stated, but should be more clearly stated as speculation or hypothesis.

3. The Discussion paragraph brings in many additional and potentially interesting pieces of evidence supporting a speculative mechanistic model. This portion is particularly interesting, but could be better defined. It would be nice to have a clearly stated model from beginning to end of what the alleles/haplotype is doing in terms of altering expression, what would be the implicated consequence of selection favoring that alteration, and how that ties in with the beneficial and pathological consequences of VEGF-A expression in hypoxia (for example, Sherpa appear to have blunted VEGF-A expression and yet have phenotypes such as increased capillary density which might be expected to require increased levels of VEGF-A). If permissible, a figure might help clarify the model being described by the authors.

4. The "Conclusions" section at lines 45-47 states "This hypoxia-tolerant VEGF-A was significantly associated with the distinctive genetic variants of the VEGFA in Sherpa highlanders." yet I see no evidence statistical evidence of significant association between VEGF-A levels and genetic variants within the Sherpa population.

Additional comments

I think one of the most interesting portions of this paper is the perplexing differences in correlation between VEGF-A concentration and SpO2 between Sherpa and non-Sherpa populations. I would be very interested to see the authors focus more on this in the results, discussion, and overall model proposed as it seems to me a unique and compelling finding.

---

## Round 0.2 · Major Revisions

Thank you for submitting a revised manuscript and for thoroughly addressing the reviewers’ comments. Two of the three reviewers were satisfied with your responses, however one of reviewers had a number of additional comments that need to be addressed. One of the comments has to do with improving the English language in your manuscript. Secondly, the reviewer suggested that the phrasing “hypoxia-tolerant” is confusing and should be replaced with “blunting hypoxia” to better clarify the fact that there is no mechanistic or structural difference in the VEGF-A protein that confers resistance to hypoxia between the populations you studied. The reviewer also wanted you to further explore the relationship between allele differences and selection, in order to be able to support any statements that differences in allele frequencies between lowlanders and Sherpas are due to natural selection. As per the reviewer’s suggestion, this includes running a formal selection test. Lastly, the reviewer suggested that you further investigate the association between the lowlander haplotype and plasma levels of VEGF-A expression or levels of VEGF-A mRNA.

Please, submit a detailed rebuttal which shows where and how you have taken all comments and suggestions into consideration. If you do not agree with some of the reviewers’ comments or suggestions, please explain why. Your rebuttal will be critical in making a final decision on your manuscript. Please, note also that your revised version may enter a new round of review by the same or by different reviewers. Therefore, I cannot guarantee that your manuscript will eventually be accepted.

Reviewer 1 ·

Basic reporting

No comment

Experimental design

No comment

Validity of the findings

No comment

Additional comments

The authors have addressed all my concerns. The manuscript is now ready for publication.

·

Basic reporting

No comment

Experimental design

No comment

Validity of the findings

No comment

Additional comments

The authors have responded adequately to my previous comments, and those of the other reviewers.

Reviewer 3 ·

Basic reporting

1. I think the word choice 'hypoxia-tolerant' is extremely confusing in this context particularly when used to describe the VEGF-A protein. The model the authors are suggesting is that SNPs in TFBS's at the promoter of VEGFA obfuscate HIF binding to some extent, ultimately down-regulating VEGFA and blunting it's response to hypoxia. In this context I think using the term blunting makes a lot more sense because the VEGF-A protein is no different from that of lowlanders and doesn't have some sort of mechanical tolerance to hypoxia as the title suggests. Rather transcript accumulation may be blunted by these alleles thus preventing an overabundance of the protein at elevation.


2. There are still a lot of instances of poor syntax, grammar, and English in general. Several examples are below:
[41-42] "locating" rather than "located" is used.
[47] "showing a character of tolerance to hypoxia"
[95-95] "Recently, an additional mechanism concerning their adaptation to hypoxia was proposed, assuming that the microcirculatory blood flow and capillary density were increased in Sherpa highlanders for promoting oxygen supply in tissues exposed to high-altitude hypoxia" - This is just an extremely convoluted sentence thats hard to parse
[153-156] "With exposure to hypoxia, VEGF-A, as a key mediator, involves in downstream productions in the hypoxia-associated biological pathway and upregulates vascular smooth muscle and endothelial cells induced by hypoxia in vivo and in vitro" - 'Involves in' is not grammatically correct. Downstream productions of what? 'Upregulating' cell types doesn't really make sense grammatically. A key mediator of what?
[662-664] "Taking all these together, it is suggested that the genetic variants of VEGFA in Himalayan highlanders are indeed characteristic among populations globally, probably by natural selection for the adaptation to the high-altitude environment" - unclear what characteristic among is meant to mean here. What is characteristic? Not the Himalayan haplotype based on previous sentence.

Experimental design

The authors have made efforts to acknowledge the flaws in their experimental design which prohibit conclusively identifying differences in the Sherpa due to biology.

Validity of the findings

Ultimately I think there is still a big problem here with inferring selection simply because the allele frequencies in this region differ significantly from lowlanders. It would be useful to know what the typical genetic divergence is in other regions of the genome to infer if this degree of divergence is extreme, also, adding an outgroup (ie using a PBS-type method for inferring selection) would be useful in understanding whether the shift in allele frequency occurred in the Sherpa lineage or the lowlander lineage. The lack of a formal selection test of any kind, and the relatively small changes in frequency between populations likely means these alleles do not show statistical signatures of selection - and indeed, the largest selection scan of Tibetan populations that I am aware of (Yang et al., 2017, PNAS) fails to turn up any selection signature at these alleles. Given that, it's harder to conclusively believe that these differences are "probably caused by natural selection" as stated or implied multiple times by the authors.

There is also an issue with the conclusions stated in the title of the article, that hypoxia-tolerant VEGFA is 'associated' with these genetic variants in Sherpa highlanders. While it is true that several SNPs have been associated with VEGFA expression in other works, there is no evaluation of these alleles statistical association with either plasma VEGF-A or VEGFA mRNA in Sherpa highlanders. Do Sherpa with the 'lowlander haplotype' have higher VEGF-A in their plasma than those with the Sherpa haplotype?

According to table 4, 15.7% of Sherpa have the most common lowlander haplotype, and 28% of lowlanders have the presumptively selected Sherpa haplotype. While the authors do note that "Expectedly, among those of individuals carrying the C-T-G-C haplotype, there was no significant difference in the VEGF-A concentrations between the Sherpas at high altitude and non-Sherpas at low altitude" [515-517], this is also the case when comparing all Sherpa highlanders to all non-Sherpa lowlanders. It is thus unclear what subsetting by this haplotype tell us. We need to know if either there are difference within groups between haplotypes or between groups for all haplotype (ie do Sherpa with the lowlander haplotype have elevated VEGF compared to lowlanders as would be predicted by the authors model). Without this information, the claim of association is conjecture based on publicly available data, not a new finding.

Additional comments

There are two novel pieces of evidence presented in this paper: 1) VEGF-A plasma concentrations in Sherpa at high altitude and in non-Sherpa lowlanders at 1300m in Nepal. 2) An evaluation of the allele and haplotype frequency for known or suspected eQTLs of VEGFA in the 5' and promoter region of the gene in these same individuals. While these two pieces of data are interesting and suggestive of a mechanistic model for blunted VEGF response to hypoxic stress in Sherpa populations, they do not, in my opinion contribute enough new data or data analysis to constitute a full paper.

---

## Round 0.3 · Major Revisions

Thank you for submitting a revised manuscript and for addressing the reviewers’ comments. Even though you have already gone through two rounds of review, there are still some lingering criticisms and comments that need to be addressed. One of the issues raised by the reviewers is that the limitations of your study need to be more clearly and thoroughly acknowledged and addressed. One of the reviewers gave detailed comments regarding lines 243-251 in your revised manuscript and how your statements can be revised. Another important point raised by the reviewers is regarding overinterpretation of your findings and toning down any population genetics claims and statements about natural selection, especially since you did not perform any of the statistical tests requested during the last round of reviews.

Please, submit a detailed rebuttal which shows where and how you have taken all comments and suggestions into consideration. If you do not agree with some of the reviewers’ comments or suggestions, please explain why. Your rebuttal will be critical in making a final decision on your manuscript. Please, note also that your revised version may enter a new round of review by the same or by different reviewers. Therefore, I cannot guarantee that your manuscript will eventually be accepted.

Reviewer 3 ·

Basic reporting

See below

Experimental design

See below

Validity of the findings

See below

Additional comments

I find this paper fundamentally flawed for the following reasons:

1. The authors cannot say anything about a difference in response in vegf for the sherpa without baseline low altitude vegf values for Sherpa individuals. Sherpa may have lower vegf at low altitude and have a normal response to hypoxia such that at high-altitude levels of vegf look similar to another populations low-altitude levels. Both cases are interesting, and i think it's ok to speculate that the Sherpa have a blunted response, but the data cannot show this. That said, I think careful wording can avoid this issue.

2. Much more significantly I am deeply concerned about the claims about natural selection and outright refusal to perform any commonly used and accepted statistical tests of selection. I have brought this comment up in both of my reviews and the authors responses are telling. In response to repeated requests for a statistical selection test that is typically used in these type of studies they replied, "we have no knowledge about the “formal selection test” as we did not find any “formal selection test” in the relevant studies on high-altitude adaptation in humans"... This response is a bit unnerving given that they cite multiple papers which use selection scans of various types in their manuscript. In response to my suggestion to use PBS as a test, the authors replied, "However, we are afraid to say that we might not perform this analysis as we could not have “an outgroup” in our study and we did not have raw data of the VEGFA SNPs in other ethnic populations." This response makes no sense as they clearly have data on global population genotyping data which they use in Fig 3. It also fundamentally misunderstands how a test like population branch statistic works. These issues could be mitigated by involving a co-author or collaborator with knowledge of population genetics, rather than making wild claims about selection while simultaneously refusing to learn how to test for it in a rigorous way. Claiming that differences in allele frequencies is a basis for claims of natural selection is not scientific or supported by the literature. Any two populations will differ in allele frequencies at any number of genes in based on genetic drift alone and neutral evolutionary processes. Just because one population lives in an extreme environment is not evidence that any differences in allele frequency they have compared to other populations are by definition a result of selective pressures.

3. Finally, the authors claim in the title of their paper that these snps in vegf are associated with the blunted response to hypoxia in Sherpa highlanders and yet they never show this. They show that the Sherpa have high frequency of alleles that have been shown in other papers to be eQTLs of vegf, they have sherpa with both the high and low altitude haplotypes of the vegf alleles, and yet they never test directly for an association within the sherpa population (ie do sherpa with the low-altitude haplotype have significantly higher vegf concentrations than those with the more common high-altitude haplotype). I have mentioned this in my comments and the authors should have the ability to do so based on the fact they have 15% of Sherpa with the common low-altitude haplotype (table 4). But they have not done so.

·

Basic reporting

See attached document entitled "Manuscript review, Peer J"

Experimental design

See attached document entitled "Manuscript review, Peer J"

Validity of the findings

See attached document entitled "Manuscript review, Peer J"

Additional comments

See attached document entitled "Manuscript review, Peer J"

---

## Round 0.4 · accepted · Accept

Thank you for thoroughly addressing the reviewers' comments and thus greatly improving your manuscript.

·

Basic reporting

No further comments.

Experimental design

No further comments.

Validity of the findings

No further comments.

Additional comments

No further comments.

·

Basic reporting

The authors have adequately responded to my criticisms of the previous version of this manuscript and/or made the requisite changes.

Experimental design

no comment (beyond those made earlier)

Validity of the findings

no comment

Additional comments

no comment